# Soil fertility management for yield and income stability in soybean-wheat rotation system under climate variability in subtropical Central India

Shinogi K. C.[1], Sanjay Srivastava📧[1]*, Radha T. K.[2], D. L. N. Rao[1¤], Bharat Prakash Meena[1], Nishant Kumar Sinha[1], Hiranmoy Das[3], Rashmi I.[4], Rosin K. G.[5], Amar Bahadur Singh[1¤], Sanjib Kumar Behera[1], Monoranjan Mohanty[1]

**1** ICAR-Indian Institute of Soil Science, Bhopal, Madhya Pradesh, India, **2** ICAR-Indian Institute of Horticultural Research, Bangaluru, Karnataka, India, **3** ICAR-Indian Institute of Vegetable Research, Varanasi, Uttar Pradesh, India, **4** ICAR-Indian Institute of Soil and Water Conservation RC Kota, Rajasthan, India, **5** ICAR-Indian Agricultural research Institute, New Delhi, India

¤ Current address: Retired, Bhopal, Madhya Pradesh, India
* sanjaysrivastava238@gmail.com

## Abstract

The soybean–wheat rotation system forms a crucial component of Central India's rainfed agriculture. Although historically productive and profitable, the system's sustainability is increasingly threatened by suboptimal nutrient management, and soil degradation under changing climate. This three-year study, conducted on small and semi-medium farms in Central India using a Participatory Technology Development (PTD) approach, aimed to evaluate how diverse nutrient management strategies influence yield, profitability, and resource use efficiency. Four nutrient management strategies were tested on small to semi-medium farms: ($T_1$) Integrated Nutrient Management (INM) using chemical fertilizers, farmyard manure and biofertilizers, ($T_2$) Integrated Nutrient Management using chemical fertilizers, enriched compost and biofertilizers, ($T_3$) Soil test crop response-based fertilization, and ($T_4$) Farmers' conventional practice. Among the four treatments $T_2$ achieved the highest system productivity (6.63 t ha⁻¹ wheat equivalent), net returns (67,680 INR ha⁻¹), and superior yield stability, under climatic variability. $T_3$ offered moderate productivity while promoting balanced nutrient use and lower input costs. Both $T_2$ and $T_3$ enhanced microbial activity and soil health, while $T_4$ led to nutrient depletion, (especially potassium), and low returns. Furthermore, the timing and distribution of monsoon rainfall significantly impacted performance and yield stability of soybean. This study highlights the critical need for integrated, site-specific, and farmer-centric nutrient management strategies to ensure sustainable productivity, profitability, and soil health under changing climatic conditions.

**Data availability statement:** All relevant data are within the paper and its Supporting Information files.

**Funding:** The author(s) received no specific funding for this work.

**Competing interests:** The authors have declared that no competing interests exist.

## Introduction

Crop diversification through legume–cereal rotation is an effective soil fertility management technique practiced globally. As a low-cost and sustainable strategy, this rotation supports both ecological intensification and economic resilience in diverse agroecosystems, particularly for smallholder and rainfed agriculture systems [1,2]. Legumes contribute nitrogen (N)-rich residues with a low carbon-to-nitrogen (C: N) ratio, which significantly enhances the yield of subsequent cereal crops [3,4]. Additionally, alternating legumes with cereals is regarded as a natural method of biofortification, improving the nutritional value of cereal grains [5].

Globally, legume–cereal rotation systems exhibit wide global diversity, reflecting variations in land suitability and the climatic adaptability of different legumes and cereals. According to Leff et al. [6], wheat (*Triticum aestivum*) is the most widely cultivated cereal, while soybean (*Glycine max*) is the leading leguminous oilseed crop globally. The soybean–wheat rotation system is especially predominant in temperate and subtropical regions [7], owing to its high productivity, profitability, and compatibility with regional climate conditions.

In India, the soybean–wheat rotation system has been practiced on the black soils (Vertisols) of Central India since the early 1980s, following the introduction of soybean as a rainfed crop in Madhya Pradesh. This rotation system was initially highly profitable, contributing significantly to regional agricultural incomes [8,9,10]. However, in recent years, climate change has severely disrupted its stability. Increasingly erratic rainfall patterns, drought stress during critical reproductive stages, and the emergence of new pests and diseases have severely impacted soybean yields. The national average soybean yield has declined to around 1.0 t ha$^{-1}$, which is significantly below the global average of 2.76 t ha$^{-1}$ [GoI, 2023, USDA, 2024].

Concurrently with climate change, adoption of higher seed rates than recommended and unscientific nutrient management practices have further reduced the profitability of soybean-wheat rotation system in India, despite wheat maintaining higher yields [Sharma, 2016]. Furthermore, in-situ burning of mechanically harvested wheat residues before summer ploughing has linked the soybean-wheat rotation system to environmental degradation. This practice deteriorates the soil by depleting organic matter and nutrients present in crop residues and destroying beneficial soil microorganisms [11,12,13].

To enhance the sustainability of the soybean-wheat rotation system in India, a range of agricultural technologies have been integrated over the years, particularly to build system resilience by improving soil health to withstand biotic and abiotic stresses, and by promoting stress-tolerant crop varieties. In line with the findings of long-term fertilizer experiments by [14] in the soybean-wheat system, soil researchers have developed precise nutrient management practices. These include integrating organic and inorganic fertilizers, retention of crop stubble in the field either as soil cover or through soil incorporation, and the recycling of stubble into nutrient-rich composts [12,10]. Additionally, to facilitate minimal soil disturbance during soybean sowing while retaining wheat stubble, customized machinery has been introduced to farmers [15].

Despite substantial efforts by researchers and government agencies, studies have shown that field-level adoption of many of the recommended practices for soybean-based systems remains insufficient to achieve their intended ecological and economic benefits [16,17].

Consequently, social researchers note that farmers' adoption decisions are not solely driven by economic profitability. Instead, they are significantly influenced by factors such as the time, labour, and effort required to implement the practices, as well as the perceived practicality and utility of the technologies within the specific context of individual farming systems. For example, in the case of crop residue incorporation, a farmer's environmental awareness and willingness to incorporate straw into their fields are key factors influencing adoption decisions [18,19].

A pilot study conducted by the ICAR-Indian Institute of Soil Science to examine the diffusion and adoption of recommended nutrient management practices in the soybean-wheat rotations revealed wide variability in the on-farm resource base of smallholder farmers. It also underscored the critical role of farmer participation in the development and evaluation of agricultural technologies. Recognizing these complexities, a study was undertaken using a Participatory Technology Development (PTD) approach in the soybean–wheat rotation system of Madhya Pradesh for a period of three years. This research aimed to determine how nutrient management practices and the efficiency with which farmers utilize available on-farm resources significantly influence yield and income in the soybean–wheat rotation system under varying environmental conditions in India.

## Materials and methods

### Study area, experimental design and treatments

The study was conducted in the Berasia region, situated in Bhopal district, Madhya Pradesh, India. Geographically, the region is located at 23.6830253°N and 77.4773669°E, with an elevation of 470 meters at the eastern edge of the Malwa Plateau in central India. The climate is classified as subtropical, with most of the rainfall occurring during the southwest monsoon season. The mean annual temperature is 25°C, and annual rainfall ranges between 850 mm and 1800 mm. Predominant cropping systems in the area include soybean–wheat and soybean–chickpea rotations, with maize and rice cultivated as rainfed crops in certain pockets. Most farmers belong to marginal to medium landholding categories [20].

An initial survey revealed low economic profitability attributed to suboptimal nutrient management practices. Farmers typically applied only nitrogenous and phosphatic fertilizers, with farmyard manure (FYM) applications limited to 1.5–2 t ha$^{-1}$ every 2–3 years based on availability. Although high yielding varieties such as soybean JS-355 (yield potential ~1.89 t ha$^{-1}$) and wheat Lok 1 and HI8498 (*Malav Shakti*) (yield potential >4.5 t ha$^{-1}$) were adopted by the farmers crop yields were only 1.2 t ha$^{-1}$ for soybean and 2.5 t ha$^{-1}$ for wheat, which clearly indicated inadequate nutrient management. To identify optimal nutrient management for maximizing yield and economic returns, Participatory Technology Development (PTD) approach was implemented across nine selected farms. Selection criteria for farmers included managerial capability and willingness to allocate resources and support trials. Farms were also selected in such a way that they equally represented landholding categories small (1–2 ha), medium (2–4 ha), and semi-medium (4–10 ha) categories.

For the field trials, four nutrient management practices were defined for the soybean-wheat crop rotation system as follows.

1. T$_1$ (INM$_{FYM}$): Integrated Nutrient Management (INM) with recommended 5 t ha$^{-1}$ FYM, *Rhizobium* biofertilizers (R33 & R34, 4 kg ha$^{-1}$), and 50% NPK for soybean; PGPR biofertilizers (P3 & P10, 4 kg ha$^{-1}$) and 75% NPK for wheat.

2. T$_2$ (INM$_{EC}$): INM developed in a participatory mode integrating enriched compost (phospho-sulpho-nitro-compost) at 2 t ha$^{-1}$, substituting 5 FYM t ha$^{-1}$ of the T$_1$ (INM$_{FYM}$), promoting on-farm residue recycling using wheat straw.

3. T$_3$ (STCR): Soil Test Crop Response inorganic input recommendations developed for the region through All India Coordinated Research Project on Soil Test Crop Response correlation (ICAR). Nutrient recommendations were calculated for each of the nine farm plots based on their initial soil test values to achieve specific target yields.

4. T$_4$ (FP): Farmers' practice involving 1 t ha$^{-1}$ FYM with nitrogenous and phosphatic fertilizers.

The Recommended Dose of Fertilizer (RDF) used as the baseline for the INM treatments ($T_1$ and $T_2$) was: 20 kg ha$^{-1}$ N, 60 kg ha$^{-1}$ $P_2O_5$ and 20 kg ha$^{-1}$ $K_2O$ for soybean crop and 120 kg ha$^{-1}$ N, 60 kg ha$^{-1}$ $P_2O_5$ and 40 kg ha$^{-1}$ $K_2O$ for wheat crop. Biofertilizers were applied in dry and liquid forms through seed treatments. The experiment followed a Randomized Block Design (RBD) in an on-farm setting, with each of the nine selected farmers serving as a replication. Treatment plots were large (1000 m² each), leading to a total evaluated area of 3.6 hectares.

## Climatic parameters

Climatic factors influencing crop growth were monitored over the three-year study during the *Kharif* (June–July to October–November) and *Rabi* (October–November to March–April) seasons. Parameters such as maximum temperature, minimum temperature, average daily temperature and rainfall amount and distribution were monitored, particularly during the critical growth stages of these crops. These data were then subjected to correlation analysis with crop yields to evaluate the influence of weather conditions on overall productivity.

## Soil analysis

Surface soil samples (0–15 cm depth) were collected pre- trial (before the first crop season)- and post-trial (after three years) from the nine farm fields. Initial composite soil samples were derived from five points per 4000 m² area to established baseline conditions. The soil samples taken pre-trial from each treatment plot were analysed for the physical and chemical properties.

Initial soil properties for the selected agroecosystem averaged as 45% sand, 13.5% silt and 41.5% clay content (Loam texture) with pH of 7.63, Electricity Conductivity (EC) of 0.2 ds m$^{-1}$, organic carbon – 6.3 g kg$^{-1}$, available nitrogen – 204.9 kg ha$^{-1}$, available phosphorus – 12.3 kg ha$^{-1}$, available potassium – 269 kg ha$^{-1}$. Specific nutrient recommendations for $T_3$ were based on these initial soil test values.

## Intercultural operations

A cost-sharing approach was adopted to facilitate the field evaluation, with the study covering the cost of new inputs (seeds, chemical and biofertilizers, compost preparation materials (portable beds, rock phosphate, iron pyrite], and plant protection chemicals). Farmyard Manure (FYM) for $T_1$ and $T_4$ was sourced on-farm.

The soybean variety used was the early maturing JS 95–60 (average yield ~1.6 t ha$^{-1}$), and the wheat variety GW-322 (average yield ~4.5 t ha$^{-1}$ even under late sowing). Seed rates were 80 kg ha$^{-1}$ for soybean and 100 kg ha$^{-1}$ for wheat for all treatments. Soybean sowing occurred mid-June to early July, contingent on monsoon onset, with wheat sowing 3–4 weeks post-soybean harvest in October. Wheat irrigation frequency varied based on water availability (3–5 irrigations). Soybean received basal NPK fertilizers; wheat received 50% basal nitrogen, full P and K, with remaining N top-dressed in two split doses along with irrigation. For the wheat crop, quantity and frequency of irrigations was based on water availability in the on-farm water resources, even though the crop received at least three irrigations in season. Herbicide (Imazethapyr 10% SL, 750 ml ha$^{-1}$) and pesticide (Triazophos 40 EC, 250 gm ha$^{-1}$ AI) were applied to soybean crop as needed.

## Total system productivity and economic analysis

Yields of grain and straw for both crops were recorded when crop plants attained physiological maturity. For yield estimation, one random 4 m² quadrat sample was collected from each treatment plot across the nine farms, and the resulting yields were extrapolated to kg ha$^{-1}$. The total system productivity of the soybean–wheat cropping system was evaluated by calculating the system yield. This involved converting the soybean yield into Wheat Equivalent Yield (WEY) using the respective annual market prices of the two crops. The wheat equivalent yield of soybean was then summed with the

actual wheat yield to derive the overall system yield. An economic analysis was conducted to determine the gross returns and total production costs, deriving a benefit-cost ratio. Production costs accounted for machinery use, seeds, manures and fertilizers, herbicides, and plant protection chemicals. Input prices and Minimum Support Prices (MSP) for soybean and wheat were tracked annually to assess farm-level economics. To evaluate the yield stability and consistency of the soybean–wheat system under different nutrient management practices, the Sustainable Yield Index (SYI) as proposed by Singh *et al.* (1990) was used.

$$SYI = (\bar{Y} - \sigma)/Ymax$$

Where: $\bar{Y}$ = Mean yield over the years, **σ** = Standard deviation (a measure of risk or yield variability) and $Y_{max}$ = Maximum observed yield. Here, low values of standard deviation and higher value of SYI indicates that the system is more sustainable.

### Nutrient balance calculation

Nutrient balances were computed for major nutrients (NPK), considering inputs and outputs in kg ha$^{-1}$ year$^{-1}$. Negative balances indicated nutrient depletion, while positive balances indicated accumulation. The average annual nutrient addition to the soybean-wheat system through manures and fertilizers under different nutrient management practices are given in Table 1.

### Statistical analysis

Data collected over three crop seasons were subjected to Combined Analysis of Variance (ANOVA) appropriate to the experimental design for various agronomic and economic parameters. A pooled (combined) analysis was conducted for yield and economic indicators to evaluate treatment effects across years. Treatment means were compared using Tukey's Honest Significant Difference (HSD) test at a significance level of $P < 0.05$. All statistical analyses were performed using SAS software (Version 9.3), employing the Generalized Linear Model procedure (PROC GLM).

## Results and discussion

### Climatic parameters

Climatic data recorded over the three years (Fig 1) indicated a distinct unimodal rainfall pattern characteristic of the region, with peak precipitation consistently occurring in July, ranging between 500–600 mm. This high-rainfall period (June to September) coincided with elevated maximum temperatures (30–36°C), during the *Kharif* season. Since the soybean variety used for the study was an early maturing one, duration of the crop was less than 100 days. Sowing was carried out after the effective onset of the monsoon (6–7 initial rainy days).

Temperature in the first *Kharif* season showed average seasonal temperature ($T_{avg}$) of 27°C with maximum ($T_{max}$) and minimum ($T_{min}$) monthly average of 33.6°C in June and 22.1°C in September, respectively. The monsoon starts in the region

**Table 1. Average annual nutrient addition to soybean-wheat system under the four treatments.**

| Treatments | Average nutrient addition in each crop season (kgha$^{-1}$) | | | | | | |
|---|---|---|---|---|---|---|---|
| | Organic manure | | | | Inorganic fertilizers | | |
| | N | P$_2$O$_5$ | K$_2$O | S | N | P$_2$O$_5$ | K$_2$O |
| **T$_1$** (INM$_{FYM}$) | 25.0 | 10.0 | 25.0 | 0.0 | 100.0 | 75.0 | 40.0 |
| **T$_2$** (INM$_{EC}$) | 20.0 | 40.0 | 42.0 | 34.0 | 100.0 | 75.0 | 40.0 |
| **T$_3$** (STCR) | 0.0 | 0.0 | 0.0 | 0.0 | 156.3 | 188.0 | 91.6 |
| **T$_4$** (FP) | 5.0 | 2.0 | 5.0 | 0.0 | 100.0 | 145.0 | 0.0 |

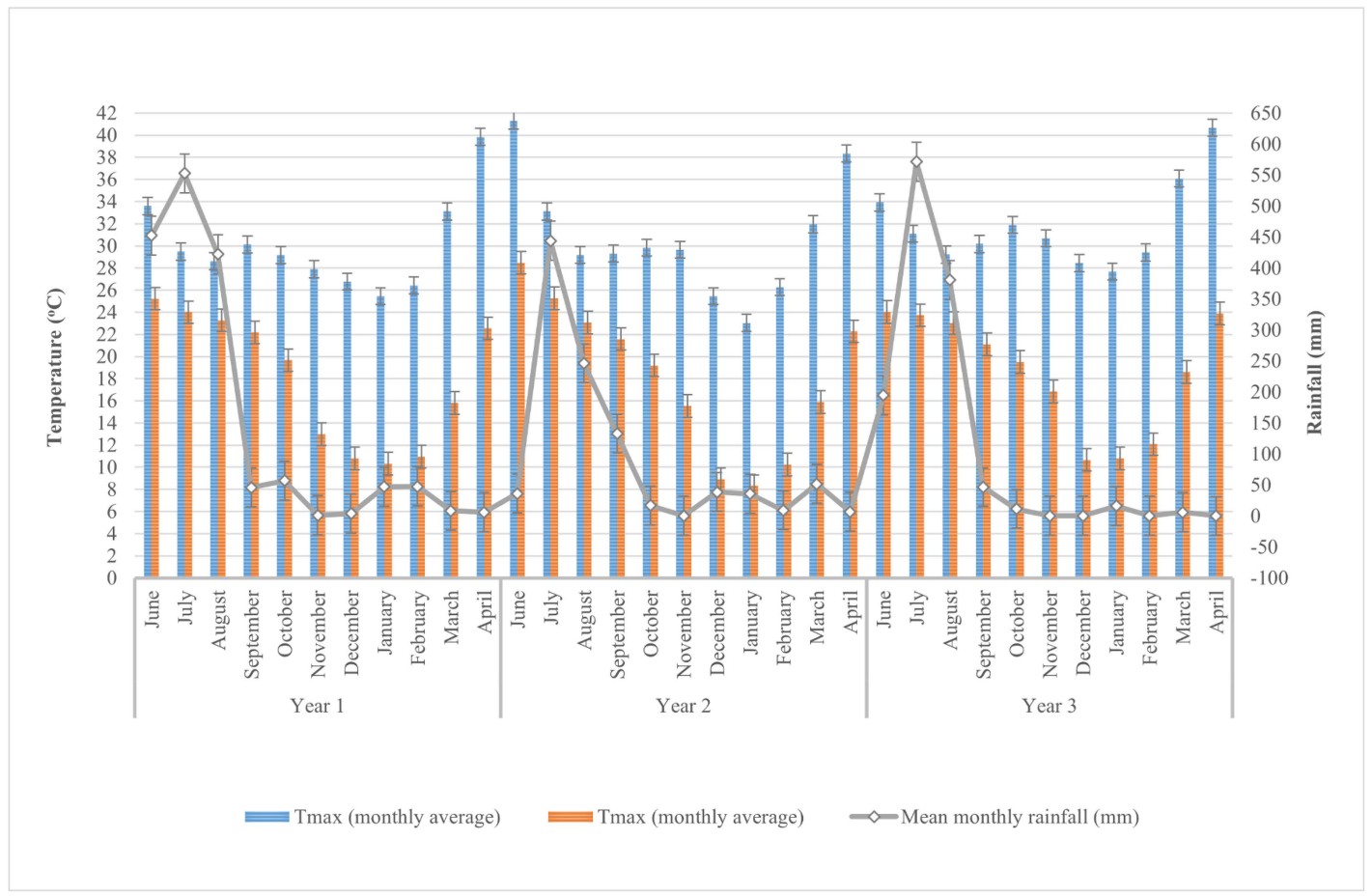

**Fig 1. Climate Graph of all the three soybean-wheat crop rotation seasons during the study period.**

by the second week of June recorded 1473.0 mm total seasonal rainfall distributed over 76 rainy days. However, the second *Kharif* season observed a slight change in the weather pattern due to the late monsoon arrival by the second week of July. The season recorded a $T_{avg}$ value of 28.9°C, $T_{max}$ of 33.1°C in the month of July and $T_{min}$ of 19.2°C in the month of October. Moreover, the season received a weak rainfall of 858.7 mm distributed over 59 rainy days. Although the third *Kharif* season followed a similar temperature pattern as that of the first season with a $T_{avg}$, $T_{max}$, $T_{min}$ values of 27.7°C, 33.9°C, and 21.1°C respectively. The monsoon was not evenly distributed with a total seasonal rainfall of 1192.9 mm spread over a short period of 51 rainy days.

The graph shows that from October onward, both rainfall and temperatures began to decline in all the three seasons, with the *Rabi* season (October to April) characterized by minimal rainfall and cooler temperatures, especially in December and January when minimum temperatures dropped to 4– to 6°C. During the first *Rabi* season the average ($T_{avg}$) maximum ($T_{max}$) and minimum ($T_{min}$) were 22.3°C, 29.2°C (October), and 10.3°C (January), respectively. During the second season the same were recorded as 22.1°C, 29.5°C (November), and 8.3°C (January) 24.2°C, 31.9°C (November), and 10.6°C (December). These conditions are conducive to wheat growth, which typically benefits from dry weather and moderate temperatures during its vegetative and reproductive stages.

The southwest monsoon, which typically arrives in the Bhopal district by mid-June and continues until the end of September, plays a crucial role in ensuring water availability during the critical growth stages of the soybean crop, thereby

improving the seed yield. Although the third crop season experienced substantial rainfall from June to August, a prolonged dry spell of 12 days in September coincided with the pod-filling stage of soybean. Drought stress during the pod development and filling stages can significantly reduce yield by decreasing the number of seeds per pod and resulting in smaller seed size [21,22,23]. Since groundwater availability for irrigation is influenced by seasonal precipitation patterns [24], variability in the arrival and withdrawal of the monsoon—as well as the intensity and distribution of rainfall during the study period—may have adversely affected the growth and yield of rainfed soybean, and consequently the wheat crop, particularly on farms with limited on-farm water resources.

Overall, the variability in the arrival and withdrawal of the monsoon, along with the intensity and distribution of rainfall during the study period, likely played a notable role in influencing the productivity of the rainfed soybean crop and, consequently, the succeeding wheat crop, particularly on farms with limited on-farm water resources. The overall climatic pattern during the study period supported the successful implementation of the soybean–wheat system.

## Impact of different nutrient management practices on crop yield

The yield performance of soybean and wheat under different treatments during the three crop rotation seasons revealed significant effects of year, treatment, and their interaction on crop productivity and sustainability (Table 2).

For soybean, treatment $T_2$ ($INM_{EC}$) consistently recorded the highest yields across the three years, with a pooled average of 1.17 t ha$^{-1}$ and the highest Sustainable Yield Index (SYI) of 0.86. $T_3$ (STCR) followed closely with a pooled yield of 1.12 t ha$^{-1}$ and SYI of 0.85. Treatment $T_4$ (Farmers' Practice) recorded the lowest performance, with a pooled yield of 0.95 t ha$^{-1}$ and the lowest SYI (0.81). Although treatment differences were not significant in the first year, they became statistically significant in subsequent years and in the pooled data, indicating a growing treatment impact over time. The significant Year × Treatment interaction (p < 0.0001) suggests that treatment effects varied considerably with climatic conditions, especially during the third year which recorded comparatively lower yields.

In wheat, treatment $T_2$ again emerged as the top performer with a pooled average yield of 4.28 t ha$^{-1}$ and a high SYI of 0.95. $T_3$ also maintained strong performance with a pooled yield of 4.06 t ha$^{-1}$ and the highest SYI among all treatments at 0.96, indicating strong yield stability across years. $T_4$ had the lowest yield (3.55 t ha$^{-1}$), although it maintained a relatively high SYI of 0.93, indicating that while its productivity was lower, it was more consistent. Unlike soybean, treatment differences in wheat were significant in all individual years and the pooled analysis (p < 0.0001), highlighting a stronger response of wheat to treatment interventions. Again, the significant Year × Treatment interaction confirms that seasonal climatic conditions influenced treatment effectiveness.

Overall, the superior performance of the two nutrient management treatments $T_2$ ($INM_{EC}$) and $T_3$ (STCR) was consistent across both soybean and wheat in terms of yield and SYI. $T_2$ provided the highest overall productivity in both crops, while $T_3$ demonstrated the highest stability in wheat (SYI of 0.96). These findings underscore the importance of optimal nutrient management in maintaining yield stability under variable climatic conditions compared to the Farmers' Practice), as reflected in the SYI. These findings support the recommendation of $T_2$ as a preferred management strategy in the soybean–wheat system for achieving both productivity and resilience. Yield advantages resulting from the use of enriched compost (phospho-sulpho-nitro compost) for soybean and wheat crops have also been reported by [25,26].

## Variation in crop yield across the farm holdings

The average seed yield of soybean and wheat crops from different nutrient management practices (Fig 2 and 3) showed notable variability across the farm holdings and years. As shown in Fig 2, treatment $T_2$ consistently produced the highest average soybean seed yields across most farm sizes in all three years, recording the highest yield of 1.6 t ha$^{-1}$ in the first crop season. Soybean yields were generally higher in larger holdings (semi-medium), followed by medium and small holdings, suggesting a possible advantage of resource availability or management efficiency in larger farms. Treatment $T_4$ consistently recorded the lowest yields across all holding categories and years, highlighting its limited effectiveness. The

**Table 2. Average yield and Sustainable Yield Index (SYI) of soybean and wheat under different nutrient management practices in the soybean–wheat crop rotation system.**

| Crop | Treatments | Average Yield (t ha$^{-1}$) | | | | SYI |
|---|---|---|---|---|---|---|
| | | Year 1 | Year 2 | Year 3 | Pooled | |
| **SOYBEAN** | T$_1$ (INM$_{FYM}$) | 1.14a | 1.15a | 0.91c | 1.07b | 0.82 |
| | T$_2$ (INM$_{EC}$) | 1.25a | 1.22a | 1.05a | 1.17a | 0.86 |
| | T$_3$ (STCR) | 1.19a | 1.17a | 1.00b | 1.12ab | 0.85 |
| | T$_4$ (FP) | 1.04a | 0.98b | 0.83d | 0.95c | 0.81 |
| *Sources of variation* | | *Probability level of significance* | | | | |
| *Year* | | – | – | – | *<0.0001* | |
| *Treatment* | | *0.1345* | *0.0004* | *<0.0001* | *<0.0001* | |
| *Year x Treatment* | | – | – | – | *<0.0001* | |
| **WHEAT** | T$_1$ (INM$_{FYM}$) | 4.06a | 3.79b | 3.90c | 3.92b | 0.93 |
| | T$_2$ (INM$_{EC}$) | 4.39a | 4.20a | 4.25a | 4.28a | 0.95 |
| | T$_3$ (STCR) | 4.13a | 3.96b | 4.08b | 4.06b | 0.96 |
| | T$_4$ (FP) | 3.56b | 3.42c | 3.67d | 3.55c | 0.93 |
| *Sources of variation* | | *Probability level of significance* | | | | |
| *Year* | | – | – | – | *<0.0001* | |
| *Treatment* | | *0.0003* | *<0.0001* | *<0.0001* | *<0.0001* | |
| *Year x Treatment* | | – | – | – | *<0.0001* | |

*Means followed by the same letter/letters are not significantly different based on Tukey's Honest Significant Difference (HSD) at probability P=0.05.*

first year showed the most favourable yield performance overall, followed by second year, with a noticeable decline in the third year, likely influenced by climatic variations such as reduced rainfall or temperature extremes, as seen in the earlier climate graph.

For the wheat crop (Fig 3) the average grain yield across different farm holding sizes and treatments demonstrated consistent trends, with Treatment T$_2$ producing the highest yields across most categories. Semi-medium holdings consistently achieved the highest yields, peaking near or above 4.8 t ha$^{-1}$, indicating more efficient resource utilization and possibly better access to inputs or mechanization. Across all years, T$_4$ recorded the lowest yields, especially in small and medium holdings, reinforcing its relatively poor agronomic performance. Year-wise, the second crop season showed slightly superior overall grain yield compared to the other years, with T$_2$ maintaining a lead regardless of farm size. Notably, the difference in yields among holding types narrowed slightly in the third year, perhaps reflecting uniform environmental stress or improved management practices across holdings.

The error bars across categories in both graphs indicate a moderate variability, with greater consistency in both soybean and wheat crop yields from T$_2$, developed through a participatory technology development approach. These trends emphasize that T$_2$ is the most effective treatment for enhancing yield from the soybean-wheat rotation system irrespective of holding size likely due to the better adaptability to recommended practices. The semi-medium holdings consistently outperform smaller categories, possibly due to better input access and agronomic practices.

## Economic performance of the soybean-wheat rotation system

An integrated assessment of yield, cost of cultivation, gross and net income, and benefit-cost ratio (BCR) across four treatments in the soybean–wheat system reveals significant difference between treatment on both crop productivity and economic returns over the three-year period (Table 3).

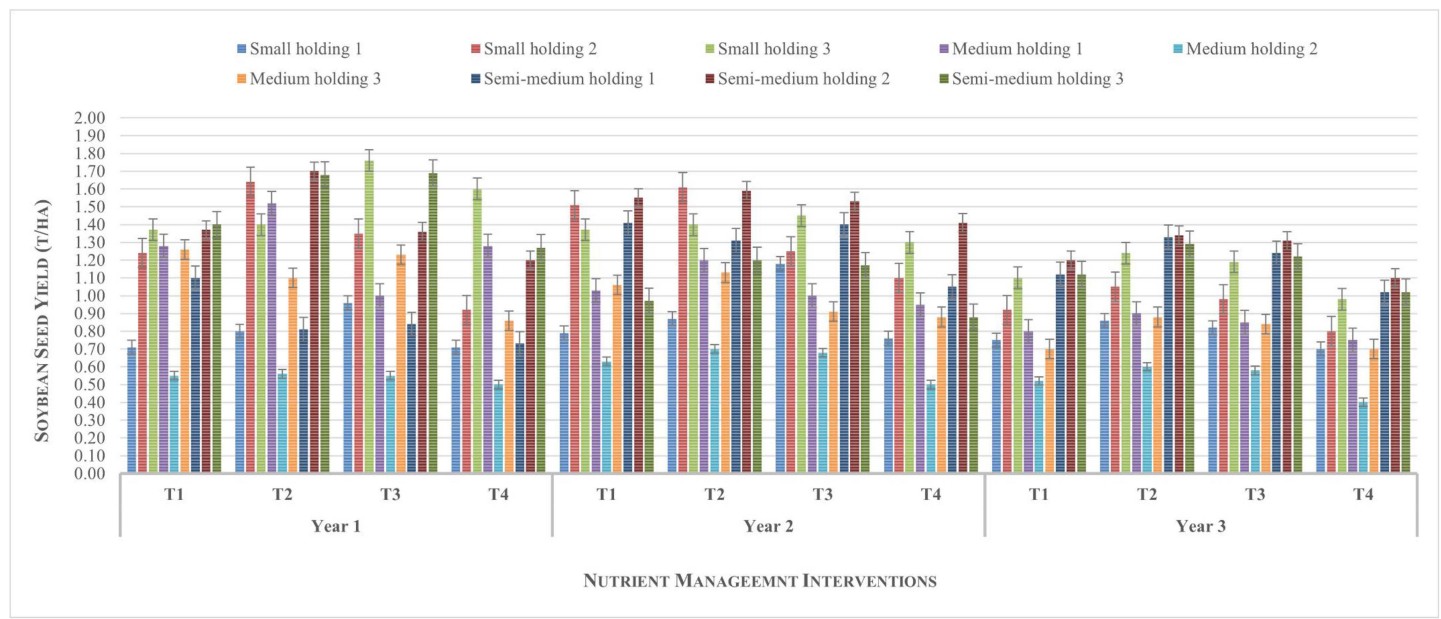

**Fig 2.  Yield of soybean across farm holdings from different nutrient management options during the study period.**

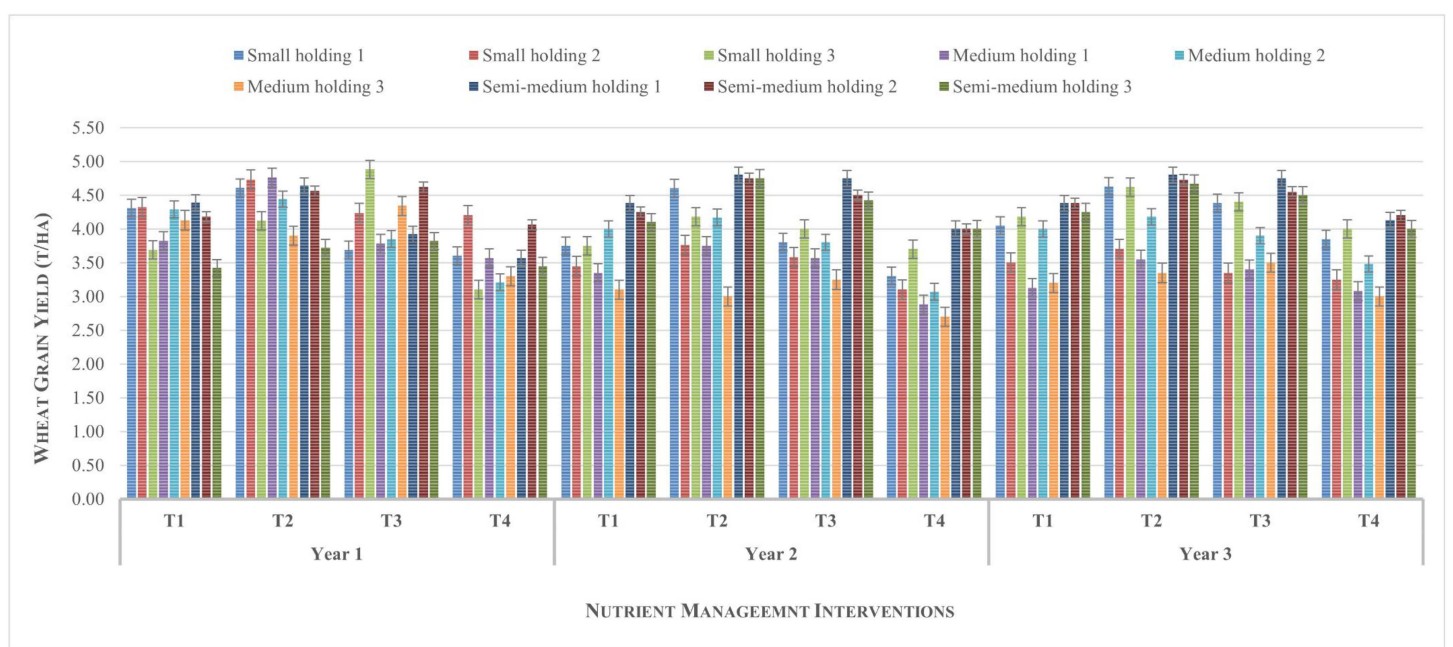

**Fig 3.  Yield of wheat across farm holdings from different nutrient management options during the study period.**

**Table 3. Economic aspects of soybean, wheat and soybean-wheat system (average of 3 years).**

| Crop | Treatments | System Yield (Pooled) | Cultivation Cost (**INR ha⁻¹) | Gross Income (INR ha⁻¹) | Net Income (INR ha⁻¹) | BCR |
|---|---|---|---|---|---|---|
| **SOYBEAN** | $T_1$ (INM$_{FYM}$) | 2.14b* | 18603.00b | 35236.67b | 16633.67b | 1.9c |
| | $T_2$ (INM$_{EC}$) | 2.35a* | 20203.00a | 38756.67a | 18553.67b | 1.9c |
| | $T_3$ (STCR) | 2.25ab* | 14650.11c | 37082.22ab | 22432.11a | 2.6a |
| | $T_4$ (FP) | 1.88c* | 14750.00c | 31313.33c | 16563.33b | 2.1b |
| *Sources of variation* | | *Probability level of significance* | | | | |
| Year | | <0.0001 | <0.0001 | <0.0001 | <0.0001 | <0.0001 |
| Treatment | | <0.0001 | <0.0001 | <0.0001 | <0.0001 | <0.0001 |
| Year x Treatment | | <0.0001 | <0.0001 | <0.0001 | <0.0001 | <0.0001 |
| **WHEAT** | $T_1$ (INM$_{FYM}$) | 3.92b | 19300.00c | 62666.67b | 43366.67b | 3.3b |
| | $T_2$ (INM$_{EC}$) | 4.28a | 19300.00c | 68426.67a | 49126.67a | 3.5a |
| | $T_3$ (STCR) | 4.06b | 25347.67a | 64911.41b | 39563.74c | 2.6c |
| | $T_4$ (FP) | 3.55c | 21750.00b | 56770.37c | 35020.37d | 2.6c |
| *Sources of variation* | | | *Probability level of significance* | | | |
| Year | | <0.0001 | 0.0035 | 0.0022 | 0.0056 | 0.0122 |
| Treatment | | <0.0001 | <0.0001 | <0.0001 | <0.0001 | <0.0001 |
| Year x Treatment | | <0.0001 | <0.0001 | <0.0001 | <0.0001 | <0.0001 |
| **SOYBEAN-WHEAT SYSTEM** | $T_1$ (INM$_{FYM}$) $T_2$ (INM$_{EC}$) $T_3$ (STCR) $T_4$ (FP) | 6.05b | 37903.00b | 97903.33c | 60000.33b | 2.6ab |
| | | 6.63a | 39503.00a | 107183.33a | 67680.33a | 2.7a |
| | | 6.30ab | 39997.78a | 101993.63b | 61995.85b | 2.6b |
| | | 5.43c | 36500.00c | 88083.70d | 51583.70c | 2.4c |
| *Sources of variation* | | *Probability level of significance* | | | | |
| Year | | <0.0001 | <0.0001 | <0.0001 | <0.0001 | <0.0001 |
| Treatment | | <0.0001 | <0.0001 | <0.0001 | <0.0001 | <0.0001 |
| Year x Treatment | | <0.0001 | <0.0001 | <0.0001 | <0.0001 | <0.0001 |

*Means followed by the same letter/letters are not significantly different based on Tukey's Honest Significant Difference (HSD) at probability P=0.05.*

**\* Wheat equivalent yield \*\* 1 INR = 0.012 USD = 0.010 EUR**

In the case of soybean, the pooled average seed yield of three years was highest for $T_2$ (1.17 t ha⁻¹), followed by $T_3$ (1.12 t ha⁻¹), with T4 recording the lowest (0.95 t ha⁻¹). Although $T_2$ recorded the highest gross income (38,756.67 INR ha⁻¹), its net income (18,553.67 INR ha⁻¹) was lower than $T_3$ due to additional cost of off-farm inputs for the preparation of enriched compost. $T_3$ emerged as the most economically efficient nutrient management practice for soybean, with the highest net income (22,432.11 INR ha⁻¹) and a superior Benefit Cost Ratio (BCR) of 2.6, owing to its relatively low cost of cultivation because of soil test-based fertilizer application. $T_4$ despite having a low yield, maintained a moderate BCR of 2.1 due to its low cultivation cost.

For the wheat crop, $T_2$ again demonstrated the highest yield (4.28 t ha⁻¹) and generated the maximum net income (49,126.67 INR ha⁻¹) with the highest BCR (3.5), highlighting its economic superiority. $T_3$, while yielding moderately (4.06 t ha⁻¹), was associated with a high cultivation cost (25,347.67 INR ha⁻¹) due to the more fertilizer requirement, which resulted in a reduced net income to 39,563.74 INR ha⁻¹ and BCR to 2.6. $T_4$ had the lowest yield (3.55 t ha⁻¹) and net income (35,020.37 INR ha⁻¹), and its BCR also remained at 2.6. This indicates that while $T_3$ was cost-intensive, its returns did not proportionally increase.

Considering the soybean–wheat rotation system, $T_2$ was the best performer with 6.63 t ha⁻¹ in terms of the overall system yield, followed by $T_3$ (6.30 t ha⁻¹), with $T_4$ recording the lowest at 5.43 t ha⁻¹. The highest gross (107,183.33 INR

ha⁻¹) and net income (67,680.33 INR ha⁻¹) were also associated with $T_2$. $T_3$ maintained competitive performance in both productivity and profitability, while $T_1$ showed moderate results. Although $T_4$ had the lowest productivity and income, it achieved a reasonable BCR of 2.4, indicating some economic viability due to lower inputs. Furthermore, the statistical analysis confirmed that year, treatment, and year×treatment interactions significantly influenced all economic and yield parameters ($p < 0.0001$). This suggests that the year-to-year variations in rainfall intensity and distribution, particularly during the monsoon months, likely contributed to the significant differences observed in system yields and economic returns across treatments.

### Impact of nutrient management practices on soil health

**Nutrient budgeting of different nutrient management systems.** The nutrient balance analysis for nitrogen (N), phosphorus (P), and potassium (K) under the soybean–wheat cropping system (Table 4) revealed significant differences across the four treatments. Treatment $T_3$ recorded the most balanced nutrient dynamics, with relatively lower negative balances for nitrogen (−19.2 kg ha⁻¹) and potassium (−19.3 kg ha⁻¹), along with the highest positive phosphorus balance (+52.9 kg ha⁻¹), indicating efficient nutrient utilization and minimal losses. In stark contrast, treatment $T_4$ showed the most severe nutrient depletion, particularly for potassium (−104.3 kg ha⁻¹) and nitrogen (−52.5 kg ha⁻¹), likely due to inadequate potassium application (only 4 kg ha⁻¹). However, $T_4$ exhibited a higher phosphorus uptake than input, it still maintained a small positive P balance.

Treatment $T_2$, despite receiving higher nutrient inputs, recorded substantial deficits in N (−42.1 kg ha⁻¹) and K (−42.7 kg ha⁻¹), suggesting either poor nutrient uptake efficiency or higher crop demand, both of which may lead to long-term soil fertility degradation if not addressed. Treatment $T_1$ showed moderate performance, with deficits in N (−17.7 kg ha⁻¹) and K (−57.0 kg ha⁻¹), and a modest P surplus (+19.4 kg ha⁻¹).

These findings reinforce that balanced nutrient application, as demonstrated in $T_3$, promotes nutrient retention and system sustainability, while imbalanced or insufficient fertilization, as in $T_4$, accelerates nutrient mining and threatens soil health. Similar concerns over nutrient mining in Indian soils have been reported by [27], who advocate for integrated nutrient management strategies. Furthermore, [28] emphasized improved nutrient uptake efficiency under soil test–based nutrient application, supporting the observed benefits in $T_3$.

**Effect of nutrient management practices on soil microbial populations.** The average on soil microbial populations (Table 5) reveals significant variations in the abundance of bacteria, fungi, and actinobacteria across the different nutrient management treatments. Bacterial populations (expressed as ×10⁷ cfu g⁻¹ dry soil) were highest under Treatment $T_3$ (2.39), which was significantly greater than $T_2$ and statistically comparable to $T_1$, indicating enhanced bacterial activity potentially due to more balanced nutrient availability in $T_3$.

In contrast, fungal populations (×10⁴ cfu g⁻¹ dry soil) peaked in Treatment $T_2$ (3.46), showing a statistically significant increase over all the other treatments. This suggests that the nutrient regime in $T_2$ was particularly favorable for fungal proliferation, possibly due to specific organic inputs or microenvironmental conditions.

**Table 4. Nutrient balance sheet of soybean-wheat crop rotation system under different nutrient management practices.**

| Treatments | Nutrient applied to soil (kg ha⁻¹) | | | Nutrient removal by crops (kg ha⁻¹) | | | Apparent Nutrient Balance (kg ha⁻¹) | | |
|---|---|---|---|---|---|---|---|---|---|
| | N | P | K | N | P | K | N | P | K |
| $T_1$ (INM$_{FYM}$) | 145 | 46 | 59 | 162.7 | 26.4 | 115.9 | −17.7 | 19.4 | −57.0 |
| $T_2$ (INM$_{EC}$) | 140 | 59 | 80 | 182.1 | 28.2 | 122.7 | −42.1 | 30.7 | −42.7 |
| $T_3$ (STCR) | 148 | 80 | 86 | 167.2 | 27.1 | 105.3 | −19.2 | 52.9 | −19.3 |
| $T_4$ (FP) | 97 | 51 | 4 | 149.5 | 31.7 | 108.3 | −52.5 | 19.3 | −104.3 |

**Table 5. Effect of different treatments on Soil microbial populations.**

| Treatments | Bacteria X10⁷ cfu g⁻¹ dry soil | Fungi X10⁴ cfu g⁻¹ dry soil | Actinobacteria X10⁵ cfu g⁻¹ dry soil |
|---|---|---|---|
| $T_1$ (INM$_{FYM}$) | 1.91ab | 2.63b | 3.01b |
| $T_2$ (INM$_{EC}$) | 1.34b | 3.46a | 3.50a |
| $T_3$ (STCR) | 2.39a | 2.09b | 2.84b |
| $T_4$ (FP) | 1.45ab | 1.30c | 2.39c |
| *Sources of variation* | *Probability level of significance* | | |
| *Treatment* | *0.0326* | *<0.0001* | *<0.0001* |

*Means followed by the same letter/letters are not significantly different based on Tukey's Honest Significant Difference (HSD) at probability P=0.05.*

Actinobacterial counts, which play a key role in organic matter decomposition and disease suppression, were highest in $T_2$ (3.50), followed by $T_1$ and $T_3$. Treatment $T_4$ consistently recorded the lowest microbial counts across all groups—fungi (1.30), bacteria, and actinobacteria (2.39)—indicating a relatively less biologically active soil environment, possibly due to imbalanced nutrient inputs or suboptimal soil conditions.

The statistically significant treatment effects ($p < 0.05$) observed across microbial groups underscore the influence of nutrient management on the soil microbial ecology. Notably, $T_2$ enhanced both fungal and actinobacterial populations, while $T_3$ was more effective in promoting bacterial abundance. These findings highlight the potential of tailored nutrient strategies to improve soil biological activity and overall soil health in the soybean–wheat cropping system.

The importance of nutrient inputs—both organic and inorganic—in modulating soil microbial diversity and function has also been emphasized in earlier studies by [29,30,31].

## Conclusion

This study examined the impact of different nutrient management practices on productivity, profitability, and soil health under the soybean–wheat rotation system of Central India, amid increasing climatic variability. The findings clearly demonstrate that nutrient management plays a pivotal role in sustaining yield and income, particularly under rainfed agriculture. Among the tested treatments, $T_2$—which combined enriched compost, biofertilizers, and chemical fertilizers—consistently delivered the highest system yield (6.63 t ha⁻¹), net income (67,680.33 INR ha⁻¹), and benefit-cost ratio (3.5) while maintaining a high sustainable yield index. This treatment also significantly enhanced fungal and actinobacterial populations, indicating improved soil biological activity. $T_3$, with soil test–based nutrient application, emerged as the most nutrient-balanced system, promoting bacterial abundance and offering competitive economic and yield performance. $T_2$ and $T_3$ proved more adaptable across diverse climatic scenarios and farm sizes, with semi-medium holdings achieving the best outcomes due to better input access [32,33,34,35,36].

Overall, the current research study emphasized on importance of integrated and balanced nutrient management in enhancing crop productivity, economic viability, and long-term soil health in the soybean–wheat system. The participatory and site-specific approaches adopted in $T_2$ and $T_3$ present scalable options for resource-limited farmers to adapt to climate change while ensuring sustainable agricultural intensification. Future research should focus on long-term nutrient cycling, water-use efficiency, and expanding these strategies to other agro-ecological regions to enhance system resilience.

## Supporting information

**S1 Data. Data underlying soybean-wheat system study.**

(XLSX)

## Acknowledgments

The authors sincerely acknowledge the support of Indian Council of Agricultural Research to carry out this research. The cooperation of farmers of the study area is also gratefully appreciated.

## Author contributions

**Conceptualization:** Shinogi K.C., Sanjay Srivastava.

**Data curation:** Hiranmoy Das.

**Formal analysis:** Bharat Prakash Meena, Nishant Kumar Sinha, Rashmi I..

**Investigation:** Shinogi K.C., Sanjay Srivastava, Radha T.K., Bharat Prakash Meena, Nishant Kumar Sinha.

**Methodology:** Shinogi K.C., Sanjay Srivastava, D.L.N. Rao, Hiranmoy Das, Rashmi I., Rosin K.G., Monoranjan Mohanty.

**Project administration:** Shinogi K.C..

**Resources:** Sanjay Srivastava, D.L.N. Rao, Amar Bahadur Singh, Sanjib Kumar Behera, Monoranjan Mohanty.

**Supervision:** D.L.N. Rao, Amar Bahadur Singh.

**Writing – original draft:** Shinogi K.C., Sanjay Srivastava.

**Writing – review & editing:** Radha T.K., D.L.N. Rao, Bharat Prakash Meena, Nishant Kumar Sinha, Hiranmoy Das, Rashmi I., Rosin K.G., Sanjib Kumar Behera, Monoranjan Mohanty.

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
