## [Decision Letter · Decision Letter 0]

5 Nov 2025

Dear Dr. Srivastava,

We look forward to receiving your revised manuscript.

Kind regards,

Abhay Omprakash Shirale, PhD

Academic Editor

PLOS ONE

Journal Requirements:

Reviewers' comments:

Reviewer's Responses to Questions

**Comments to the Author**

1. Is the manuscript technically sound, and do the data support the conclusions?

Reviewer #1: Yes

Reviewer #2: Yes

2. Has the statistical analysis been performed appropriately and rigorously?

Reviewer #1: Yes

Reviewer #2: Yes

3. Have the authors made all data underlying the findings in their manuscript fully available?

Reviewer #1: Yes

Reviewer #2: Yes

4. Is the manuscript presented in an intelligible fashion and written in standard English?

Reviewer #1: Yes

Reviewer #2: Yes

Reviewer #1: The topic is relevant to research work and all required data is included.the result and discussion part is written in a proper manner and discussed well. some grammatical mistakes have been found which is incorporated. Minor revisions are incorporated and it has been suggested in box. please do the needful.

Reviewer #2: The paper is technically sound with enough data on crop yield from consecutive years during the experiment and economic performance is also done to understand the soybean-wheat rotation system.

Please correct "diploid" from line no 144.

There are some minor typographical errors that may be corrected before uploading the revised version.

**Do you want your identity to be public for this peer review?** For information about this choice, including consent withdrawal, please see our Privacy Policy

Reviewer #1: No

Reviewer #2: **Yes:** Karthika K.S.

---

## [Author Response · Author response to Decision Letter 1]

24 Nov 2025

A detailed, point-by-point response to reviewer comments has been uploaded as a separate file titled “Response to Reviewers”.

---

## [Decision Letter · Decision Letter 1]

2 Jan 2026

Soil fertility management for yield and income stability in soybean–wheat rotation system under climate variability in subtropical Central India

PONE-D-25-53731R1

Dear Dr. Srivastava,

We’re pleased to inform you that your manuscript has been judged scientifically suitable for publication and will be formally accepted for publication once it meets all outstanding technical requirements.

Kind regards,

Abhay Omprakash Shirale, PhD

Academic Editor

PLOS One

Reviewers' comments:

Reviewer's Responses to Questions

**Comments to the Author**

Reviewer #1: All comments have been addressed

Reviewer #2: All comments have been addressed

2. Is the manuscript technically sound, and do the data support the conclusions?

Reviewer #1: Yes

Reviewer #2: Yes

3. Has the statistical analysis been performed appropriately and rigorously?

Reviewer #1: Yes

Reviewer #2: Yes

4. Have the authors made all data underlying the findings in their manuscript fully available?

Reviewer #1: Yes

Reviewer #2: Yes

5. Is the manuscript presented in an intelligible fashion and written in standard English?

Reviewer #1: Yes

Reviewer #2: Yes

Reviewer #1: the research paper is well written and now it is ready for publication after all corrections have been incorporated by author.

Reviewer #2: The paper is technically sound with appropriate methodologies involved in research.

Proper statistical analyses are also carried out and the results are discussed.

**Do you want your identity to be public for this peer review?** For information about this choice, including consent withdrawal, please see our Privacy Policy

Reviewer #1: No

Reviewer #2: No

---

## [Editor Report · Acceptance letter]

PONE-D-25-53731R1

PLOS One

Dear Dr. SRIVASTAVA,

I'm pleased to inform you that your manuscript has been deemed suitable for publication in PLOS One. Congratulations! Your manuscript is now being handed over to our production team.

Kind regards,

on behalf of

Dr. Abhay Omprakash Shirale

Academic Editor

PLOS One